# Understanding and Mitigating Gender Bias in LLMs via Interpretable Model Editing

## Abstract

Large language models (LLMs) have achieved great success in various tasks. While LLMs can learn powerful capabilities from large datasets, they also inherit the gender bias present in that data. Existing studies usually propose methods to reduce bias by data cleaning and model retraining/fine-tuning. Although these methods have shown some success, the cost of designing data and retraining/fine-tuning an LLM increases significantly as the model size grows larger. Furthermore, a lack of understanding of the mechanisms behind gender bias prevents researchers from effectively tailoring solutions to address it. In this paper, we utilize mechanistic interpretability methods to construct the neuron circuits for gender bias cases and locate the important neurons storing gender bias. Then we propose the Interpretable Model Editing (Interpret-ME) method to reduce gender bias without designing huge datasets or fine-tuning. Compared to fine-tuning methods, our approach shows competitive results in reducing gender bias across experiments with 8 LLMs. At the same time, our method does not affect the performance in other tasks. Overall, our analysis is useful for understanding the mechanism of gender bias and our method paves a potential way for reducing bias.

## 1 Introduction

Transformer-based (Vaswani et al., 2017) large language models (LLMs) (Brown et al., 2020; Ouyang et al., 2022; Chowdhery et al., 2023) have been successful in various downstream tasks. They can acquire diverse abilities from large amounts of training data. However, they also learn, perpetuate, and amplify the biases present in the data, including those related to race (Blodgett, 2021), gender (Bender et al., 2021), and religion (Li et al., 2020). If not addressed, these biases may pose unknown risks to society. Among these biases, gender bias is found to be the most ingrained and hardest to eliminate (Ranaldi et al., 2023), as it is often hidden in most sentences. Therefore, mitigating gender bias in LLMs has become an increasingly important question recently.

Although many studies have explored reducing gender bias, three main questions remain unresolved. First, many previous studies use data augmentation (Webster et al., 2020; Pant & Dadu, 2022) and data filtering (Garimella et al., 2022; Borchers et al., 2022) methods to create balanced datasets. However, this process requires substantial human resources, as the size and quality of the dataset significantly affect the LLMs' performance. Also, these methods can introduce factuality errors and bring potential risks (Kumar et al., 2022). Second, most studies update the model parameters by fine-tuning on specially designed datasets (Gira et al., 2022; Zhou et al., 2023; Ranaldi et al., 2023). However, the computational expense of fine-tuning LLMs has become a significant concern, particularly as the size of LLMs continues to increase. At the same time, Gallegos et al. (2024) point out that in-training methods risk corrupting the pre-trained language understanding due to catastrophic forgetting (Kirkpatrick et al., 2017), as fine-tuning datasets are relatively small compared to the original training data. This can impair the model's overall performance. Third, the overall localization and mechanisms by which LLMs store and produce gender bias are still unclear, preventing researchers from designing targeted mitigation solutions. Numerous studies (Zhao et al., 2023; Li et al., 2023; Gallegos et al., 2024) have pointed out that it is essential to increase LLMs' interpretability and understand which components of LLMs encode these biases.

In this paper, we propose the Interpretable Model Editing (Interpret-ME) method for reducing gender bias that addresses the aforementioned questions. Our method eliminates the need for designing

large datasets or extensive fine-tuning by focusing on editing just a few neurons (around 30 neurons) through the analysis of only ten gender bias sentences. Additionally, our Interpret-ME method is grounded in mechanistic interpretability analysis, which is helpful for understanding the mechanisms and parameter storage associated with gender bias.

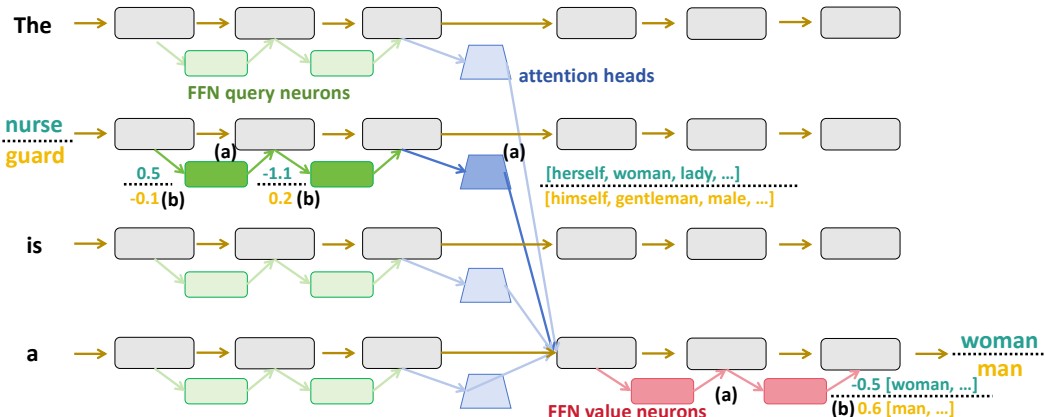

Figure 1: (a) Constructing the neuron circuits of sentence "The nurse is a => woman" and "The guard is a => man". (b) Editing the important neurons by reducing their coefficient scores.

Our method is based on the interpretability analysis of gender bias sentences. We use neuron-level interpretability methods (Yu & Ananiadou, 2023) to build the neuron circuit including shallow FFN neurons, attention heads and deep FFN neurons, as shown in Figure 1. Take "The nurse is a => woman" as an example. "nurse" activates several shallow FFN neurons useful for "increasing woman's probability and reducing man's probability". These shallow FFN neurons become interpretable after transformation of an attention head, because the FFN neurons can activate the attention neurons related to "woman". When projecting into unembedding space, these transformed FFN neurons' top tokens are related to "woman" and the last tokens are related to "man". Then the transformed FFN neurons are transferred into last position and activating the deep FFN neurons related to "woman". Furthermore, we conduct analysis on "The guard is a => man" and find that some neurons are important in both cases. The only difference is that the coefficient scores of these neurons have opposite signs. Therefore, these neurons are "gender neurons" storing gender bias.

Based on the interpretability analysis, we propose the Interpret-ME method with three stages, named locate-analyze-edit. Firstly, we only choose five gender bias cases for each gender, and then locate the important neurons in these cases. Secondly, we analyze and filter the neurons distinguishing "man" and "woman" in top/last tokens when projecting into unembedding space. Lastly, we edit these neurons by shrinking the neurons' coefficient scores. After editing these neurons, the storage of gender bias is reduced. We compare our method with state-of-the-art fine-tuning methods on 8 LLMs with parameters from 1.1B to 13B, and the experimental results demonstrate that our Interpret-ME method perform better than theirs on all the models. At the same time, we evaluate the edited models in four other common tasks and the performance does not drop compared with the original models.

Overall, our contributions are as follows:

a) We utilize interpretability methods to construct the neuron circuits of gender bias sentences and analyze the roles of different neurons. Our analysis is important for understanding the parameter storage and mechanism for LLMs to produce gender bias.

b) We propose the Interpretable Model Editing (Interpret-ME) method, which can reduce the gender bias without designing new datasets or fine-tuning stages. By only editing a few neurons, the gender bias is reduced and the performance of other tasks remains unchanged.

c) Our experiments are conducted in 8 LLMs and 3 gender bias datasets. Our ablation study explores the importance of different neurons, the number of edited neurons, and the difference of editing methods. Our work provides an important guide for neuron-level model editing. The code and data will be published on github.

## 2 RELATED WORK

### 2.1 REDUCING GENDER BIAS IN LLMS

Many studies have explored methods to reduce the gender bias in LLMs based on data selection/argumentation. Liu et al. (2021) design matched pairs to argument the training data. Ghanbarzadeh et al. (2023) generate new data by masking the gender words and predicting a new word by another language model. Zayed et al. (2023) produce a method to extract and argument the most important gender sentences. Garimella et al. (2022) and Borchers et al. (2022) design methods to filter the low-gender sentences. Han et al. (2021) and Orgad & Belinkov (2022) propose methods to compute the importance of sentences and re-weight all the sentences.

Another type of studies focus on modifying model architectures. Lauscher et al. (2021) utilize adapter modules (Houlsby et al., 2019) to mitigate gender bias. Han et al. (2021) propose a gate module to help the model take protected attributes into consideration. Also, many studies (Gaci et al., 2022; Yang et al., 2023; Woo et al., 2023) propose methods by modifying the loss functions, which can encourage the model to generate de-biasing outputs.

### 2.2 MECHANISTIC INTERPRETABILITY FOR LANGUAGE MODELS

The goal of mechanistic interpretability is to reverse engineer the internal circuit from inputs to outputs, thereby helping to understand the mechanisms of language models. Elhage et al. (2021) find that the induction heads are the main roles for predictions like [A][B]...[A] => [B]. Olsson et al. (2022) investigate the induction heads and find that these heads may be important for in-context learning. Meng et al. (2022) utilize causal mediation analysis method (Vig et al., 2020) to identify the important hidden states in GPT and find that the medium feed-forward network (FFN) layers are significant for storing factual knowledge. Geva et al. (2023) find a three-step internal mechanism for attribute extraction in factual knowledge. A common interpretability method for analyzing the internal vectors is to project them into the unembedding space Geva et al. (2022); Dar et al. (2022).

Recently, several studies try to locate the important neurons in LLMs, as numerous studies (Nanda et al.; Lieberum et al., 2023; Stolfo et al., 2023) point out finding the important neurons is of great significance for understanding the mechanism and knowledge storage in LLMs. Geva et al. (2022) find that the FFN neurons are interpretable when analyzing in unembedding space. Yu & Ananiadou (2023) propose a method to identify the deep layers' "value neurons" directly contributing to the predictions, and the shallow layers' "query neurons" contributing by activating the "value neurons".

## 3 METHODOLOGY

In this section, we first introduce the background regarding the definition of "neurons" and how to locate and analyze the important neurons in LLMs in Section 3.1. Then we conduct interpretability analysis for gender bias sentences in Section 3.2, in order to understand the parameter storage and overall mechanism of gender bias. In Section 3.3, we introduce our proposed Interpretable Model Editing (Interpret-ME) method for mitigating gender bias in LLMs.

### 3.1 BACKGROUND

**Inference pass of LLMs.** We first introduce the inference pass in decoder-only LLMs. The input sequence is $X = [x_1, x_2, ..., x_T]$ with $T$ tokens. The model generates an output distribution $Y$ (a $B$-dimension vector) over $B$ tokens in vocabulary $V$. Each token $x_i$ at position $i$ is transformed into a word embedding $h_0^i \in \mathbb{R}^d$ by the embedding matrix $E \in \mathbb{R}^{B \times d}$. Then the word embeddings are fed into $L + 1$ transformer layers ($0th - Lth$). Each layer output $h_i^l$ (layer $l$, position $i$) is computed by the sum of the previous layer output $h_i^{l-1}$, the multi-head self-attention (MHSA) layer output $A_i^l$, and the feed-forward network layer (FFN) output $F_i^l$:

$$h_i^l = h_i^{l-1} + A_i^l + F_i^l \tag{1}$$

The last layer output at the last position $h_T^L$ is utilized to calculate the final probability distribution $Y$ by multiplying the unembedding matrix $E_u \in \mathbb{R}^{B \times d}$:

$$Y = softmax(E_u h_T^L) \tag{2}$$

The MHSA output is computed by the sum of all $H$ head outputs, and each head output is an weighted sum on all positions:

$$A^l = \sum_{j=1}^{H} \sum_{p=1}^{T} \alpha_{j,p}^l \cdot O_j^l V_j^l h_p^{l-1} \tag{3}$$

where $\alpha_{j,p}^l$ is the attention score at position $p$, head $j$, layer $l$, computed by the softmax function over all positions' attention scores. $V_j^l$ and $O_j^l$ are the value matrix and output matrix in head $j$, layer $l$. The FFN output is calculated by a nonlinear $\sigma$ on two MLPs $W_{fc1}^l \in \mathbb{R}^{N \times d}$ and $W_{fc2}^l \in \mathbb{R}^{d \times N}$.

$$F_i^l = W_{fc2}^l \sigma(W_{fc1}^l(h_i^{l-1} + A_i^l)) \tag{4}$$

**Definition of neurons.** According to Geva et al. (2020), the FFN layer output can be represented as the weighted sum of many FFN subvalues:

$$F_i^l = \sum_{k=1}^{N} m_{i,k}^l fc2_k^l \tag{5}$$

$$m_{i,k}^l = \sigma(fc1_k^l \cdot (h_i^{l-1} + A_i^l)) \tag{6}$$

where the subvalue $fc2_k^l$ is the $kth$ column of $W_{fc2}^l$, and its coefficient score $m_{i,k}^l$ is based on the inner product between the residual output $(h_i^{l-1} + A_i^l)$ and the subkey $fc1_k^l$ (the $kth$ row of $W_{fc1}^l$). In this paper, we definite one neuron as the combination of the FFN subvalue and its subkey. Similar to FFN layers, the value matrix $V_j^l$ and output matrix $O_j^l$ in each attention head are also two MLPs, and the $kth$ attention neuron in head $j$, layer $l$ is definited as the combination of the attention subvalue (the $kth$ column of $O_j^l$) and the attention subkey (the $kth$ row of $V_j^l$).

**Locating and analyzing important neurons in LLMs.** Geva et al. (2022) find that the FFN subvalues are interpretable when projecting into the unembedding space. Specifically, they multiply each subvalue $v^l$ with the unembedding matrix to compute the distribution $D_{v^l}$ and analyze which tokens have the largest probabilities (top tokens) and the smallest probabilities (last tokens):

$$D_{v^l} = softmax(E_u v^l) \tag{7}$$

Based on Eq.7, Yu & Ananiadou (2023) utilize the increase of log probability of each subvalue as the importance score of FFN neurons $v_F^l$ and attention neurons $v_A^l$:

$$Imp(v_F^l) = log(p(w|v_F^l + A^l + h^{l-1})) - log(p(w|A^l + h^{l-1})) \tag{8}$$

$$Imp(v_A^l) = log(p(w|v_A^l + h^{l-1})) - log(p(w|h^{l-1})) \tag{9}$$

They name the neurons with largest scores **"value neurons"** as these neurons directly contribute to the final predictions and are distributed in deep FFN and attention layers. At the same time, there are **"query neurons"** in shallow layers, which contribute by activating the "value neurons". For every FFN neuron, they calculate the FFN neuron's query score by summing the inner products between the FFN neuron's subvalue and the subkeys of "value attention neurons". Then they sort all the FFN neurons' query scores to find the most important FFN neurons working as "query neuron".

## 3.2 NEURON CIRCUITS OF GENDER BIAS SENTENCES

Gender bias arises from the probability differences assigned to male and female terms based on the same word. Using the method introduced in the previous section, we construct the neuron circuits of gender bias sentences "The nurse is a => woman" and "The guard is a => man", in order to explore why "nurse" is more associated with "woman" and "guard" is more associated with "man". The analysis is conducted in Llama-7B (Touvron et al., 2023) with 32 layers. Each attention layer has 32 heads and each head has 4,096 neurons. Each FFN layer has 11,008 neurons.

We first analyze "The nurse is a => woman". We identify and analyze the top10 FFN value neurons, top10 attention value neurons, and top10 FFN query neurons. Both FFN value neurons and attention value neurons are interpretable. When projecting into unembedding space, many neurons' top tokens

Table 1: Top tokens and last tokens when projecting the identified value neurons into unembedding space, identified by "The nurse is a => woman".

| neuron | coeff | top tokens in unembedding space | last tokens in unembedding space |
|---|---|---|---|
| $F_{1891}^{25}$ | -1.0 | [boys, boy, Boys, Boy, men, male, guys, males, Men] | [women, ladies, Women, girls, woman, girl, Woman] |
| $F_{3114}^{20}$ | 1.1 | [herself, mother, woman, Woman, daughter, sister, mom, lady] | [himself, son, male, father, Male, brother, boy] |
| $A_{83}^{18,7}$ | 1.7 | [herself, lady, woman, actress, women, female, Woman, girl] | [himself, his, homme, mascul, mens, his, father, him] |
| $A_{54}^{18,7}$ | 1.2 | [girl, daughter, actress, woman, female, lady, Girl, females, girls] | [Men, sede, Mens, flug, gentlemen, men, Virtual, abase] |

Table 2: Top tokens and last tokens when projecting the transformation (head 7, layer 18) of identified query neurons into unembedding space, identified by "The nurse is a => woman".

| neuron | coeff | top tokens in unembedding space | last tokens in unembedding space |
|---|---|---|---|
| $F_{2026}^{4}$ | 0.7 | [herself, woman, Woman, lady, actress, women, Women, girl, she] | [himself, male, mascul, Male, gentleman, males, gentlemen, boy] |
| $F_{6772}^{16}$ | -3.5 | [himself, boys, male, 'boy', Boys, Male, mascul, males, gentleman] | [herself, woman, Woman, lady, actress, Frau, women] |

are related to "woman" and last tokens are related to "man", as shown in Table 1. $F_{1891}^{25}$ means the $1891th$ neuron in $25th$ FFN layer, and $A_{83}^{18,7}$ is the $83th$ neuron in $7th$ head, $18th$ layer. These value neurons can distinguish "woman" and "man" by increasing the top tokens' probabilities and decreasing the last tokens' probabilities when the coefficient scores are larger than 0.

Since the query FFN neurons can activate the value attention neurons, we calculate the query neurons' transformation by the important attention heads and find that the transformed vectors become interpretable like the value neurons, as shown in Table 2. Based on the results of Table 1 and 2, the neuron circuit for sentence "The nurse is a => woman" is established. In shallow layers, the query FFN neurons (such as $F_{2026}^{4}$ and $F_{6772}^{16}$) are activated by word "nurse". Then the query neurons activate several attention value neurons (such as $A_{54}^{18,7}$ and $A_{83}^{18,7}$) related to "woman", thus the transformed vectors of these query neurons can enhance "woman" probabilities and reduce "man" probabilities (Table 2). Finally, the transformed vectors are transferred into the last position and activate the FFN value neurons (such as $F_{3114}^{20}$ and $F_{1891}^{25}$).

Table 3: Important query neurons and their coefficients in "The nurse is a => woman" and "The guard is a => man". The top/last tokens are the vectors transformed by head 7, layer 18.

| neuron | coeff | top tokens in unembedding space | last tokens in unembedding space |
|---|---|---|---|
| $F_{17}^{11}$ | 0.5/-0.1 | [herself, woman, actress, Woman, lady, women, girl, femme, female] | [himself, gentleman, male, mascul, Male, males, gentlemen, boy] |
| $F_{6938}^{14}$ | -1.1/0.2 | [himself, male, gentleman, Male, mascul, males, his, boy] | [herself, woman, Woman, lady, actress, women, girl, female, femme] |

Additionally, we analyze the neuron circuit of sentence "The guard is a => man". We find two query neurons important in both sentences, as shown in Table 3. The sign of these neurons' coefficient score are different in the two cases. For instance, the coefficient score of $F_{17}^{11}$ is 0.5 activated by word "nurse", while it is -0.1 activated by "guard". This observation enhances our understanding about the mechanism of gender bias: $F_{17}^{11}$ and $F_{6938}^{14}$ stores important parameters for distinguishing "man" and "woman". When the sign of the coefficient scores changes, the neurons switch from increasing the probability of "woman" to increasing the probability of "man". We observe similar results in OPT (Zhang et al., 2022) and BLOOM (Le Scao et al., 2023), detailed in Appendix A.

## 3.3 INTERPRETABLE MODEL EDITING

Based on the interpretability analysis in Section 3.2, we hypothesize that: a) Gender bias parameters are stored in both query neurons and value neurons. b) The coefficient scores of these neurons can affect the probabilities of different genders. c) The query neurons are more important for gender bias if their transformed vectors' top/last tokens are "woman"/"man" or "man"/"woman".

According to the hypothesis, we propose the Interpretable Model Editing (Interpret-ME) method to mitigate the gender bias in LLMs. In order to identify the important neurons, we choose only 5 gender bias sentences for each gender like "The XX is a" (XX is a profession), shown in Appendix B. Our method has three steps, named **locate-analyze-edit**. First, we **locate** the topM FFN value neurons, topN attention value neurons and topP FFN query neurons for each sentence, and calculate each neuron's important score averaged on all 5 sentences for each gender. In this step, we get M+N+P neurons for "man" and M+N+P neurons for "woman". Due to superposition (Elhage et al., 2022), several query neurons not only affect gender bias performance but are also important for other tasks. Therefore, we then **analyze** and filter the query neurons whose transformed vectors' top/last tokens are opposite about "man"/"woman". Last, we **edit** these filtered neurons by shrinking their coefficient scores. Specifically, we design two editing methods, which we called zero-editing and division-editing. In zero-editing, we replace the neuron's subvalue ($fc2_k^l$ in Eq.5) with a zero vector having the same dimension. In division-editing, we divide the subvalue's each dimension by a constant score D. This method has the same result with dividing the neuron's coefficient score by D.

**Advantages.** As introduced in Section 3.3, our method only requires 10 sentences and do not require fine-tuning. The computational cost for identifying the important neurons can be done within 3 minutes in Llama-7B. Furthermore, the neuron analyzing and filtering stage helps us understand the mechanism more deeply. Hence, our method can solve the three questions mentioned in Section 1.

## 4 EXPERIMENTS AND ANALYSIS

We introduce the datasets, evaluation metrics and models in Section 4.1, and show the experimental results in Section 4.2. The ablation study and analysis are conducted in Section 4.3 and 4.4.

### 4.1 DATASETS, METRICS AND MODELS

**Datasets.** The experiments are done on StereoSet (Nadeem et al., 2020), WinoGender (Zhao et al., 2018), and Crows-Pairs (Nangia et al., 2020), which are widely used to evaluate the gender bias in LLMs (Brown et al., 2020; Ouyang et al., 2022; Touvron et al., 2023). StereoSet has 1,026 sentence pairs, each containing three sentences: a stereotype sentence, an anti-stereotype sentence, and a nonsensical sentence. WinoGender and Crows-Pairs contain 1,165 and 262 gender-bias sentence pairs, respectively, where each pair consists of two sentences with different genders.

**Metrics.** For each sentence in StereoSet, we calculate the likelihood normalized by the number of characters (Gao et al., 2021). If a sentence's normalized likelihood is the largest among the three sentences in the sentence pair, this sentence is "chosen" by the model. We follow the original StereoSet paper's metrics, including language modeling score (LMS), stereotype score (SS), normalized SS (NSS), and Idealized CAT score (ICAT). LMS is the percentage when the model chooses a logical answer (either the stereotyped or anti-stereotyped answer) over the nonsensical answer. SS represents the percentage when the model chooses the stereotyped answer over the anti-stereotyped answer. For the ideal language model, its LMS would be 100 and its SS would be 50. In this situation, the model chooses 50% stereotyped answers, 50% anti-stereotyped answers, and 0% nonsensical answers. The ICAT score is the product of LMS and Normalized SS (NSS):

$$ICAT = LMS \cdot \frac{min(SS, 100 - SS)}{50} \qquad (10)$$

For each sentence pair in WinoGender and Crows-Pairs, we compute the difference of the entropy (Brown et al., 2020) between the two sentences in each pair, named "entropy difference". If the entropy difference becomes smaller after using our method, it means that the gender bias is reduced.

**Models.** We conduct experiments in decoder-only LLMs with parameters from 1.1B to 13B including Llama (Touvron et al., 2023), OPT (Zhang et al., 2022) and BLOOM (Le Scao et al., 2023).

**Evaluation on common datasets.** We also conduct experiments to verify whether the performance on other tasks are affected after the models are edited. Similar to the evaluation datasets in modern LLMs (Brown et al., 2020; Touvron et al., 2023), we evaluate the accuracy on common datasets including PIQA (Bisk et al., 2020), ARC easy (Clark et al., 2018), RACE (Lai et al., 2017) and arithmetic (Brown et al., 2020).

## 4.2 OVERALL RESULTS

We compare our Interpret-ME method with the state-of-the-art fine-tuning method (Ranaldi et al., 2023) on StereoSet dataset. The metrics (LMS, SS, NSS, ICAT) are introduced in Section 4.1. The results are shown in Table 4. On all the models, our Interpret-ME method show a competitive result.

Table 4: LMS (larger better), SS (smaller better), NSS (larger better), ICAT (larger better) scores of StereoSet dataset on fine-tune method and our Interpret-ME method.

| | **fine-tune** | | | | **Interpret-ME** | | | |
|---|---|---|---|---|---|---|---|---|
| model | LMS ↑ | SS ↓ | NSS ↑ | ICAT ↑ | LMS ↑ | SS ↓ | NSS ↑ | ICAT ↑ |
| Llama-7B | 91.91 | 68.62 | 62.76 | 57.69 | 94.54 | 67.73 | 64.52 | 61.00 |
| Llama-13B | 92.74 | 69.59 | 60.82 | 56.40 | 95.41 | 68.51 | 62.96 | 60.08 |
| OPT-1.3B | 92.98 | 69.3 | 61.4 | 57.09 | 93.90 | 64.62 | 70.76 | 66.44 |
| OPT-2.7B | 92.54 | 68.13 | 63.74 | 58.99 | 93.90 | 65.98 | 68.03 | 63.88 |
| OPT-6.7B | 93.03 | 68.62 | 62.76 | 58.39 | 94.49 | 64.13 | 71.73 | 67.78 |
| BLOOM-1.1B | 91.76 | 65.5 | 69.00 | 63.32 | 92.64 | 65.69 | 68.61 | 63.56 |
| BLOOM-1.7B | 92.01 | 65.98 | 68.04 | 62.59 | 93.61 | 65.59 | 68.81 | 64.41 |
| BLOOM-3B | 92.25 | 68.32 | 63.36 | 58.44 | 93.32 | 65.98 | 68.03 | 63.48 |

Then we evaluate the change of entropy difference when using our Interpret-ME method on Wino-Gender (WinoG) and Crows-Pairs (CPairs). We also compute the accuracy change on common datasets including PIQA, ARC easy, RACE and arithmetic (arithm). The results are shown in Table 5. Except on OPT-6.7B WinoGender, the gender bias is mitigated in all the models and datasets, and the performance on the 4 common datasets are not affected much. Overall, the results shown in Table 4 and Table 5 can prove that our Interpret-ME method can successfully reduce the gender bias without hurting other abilities of the model much. Therefore, our Interpret-ME method provides an effective way to mitigate gender bias without designing large datasets and fine-tuning methods.

Table 5: Change of entropy difference (smaller better) on WinoG/CPairs and accuracy (larger better) on 4 common datasets (PIQA, ARC, RACE, arithm) when using Interpret-ME method.

| model | WinoG ↓ | CPairs ↓ | PIQA ↑ | ARC ↑ | RACE ↑ | arithm ↑ | avg ↑ |
|---|---|---|---|---|---|---|---|
| Llama-7B | -0.0002 | -0.0011 | +0.06% | -0.35% | +0.0% | -0.11% | -0.1% |
| Llama-13B | -0.0002 | -0.0002 | +0.1% | -0.18% | +0.0% | -0.12% | -0.05% |
| OPT-1.3B | -0.0002 | -0.0005 | -0.3% | +0.5% | -1.5% | +0.9% | -0.1% |
| OPT-2.7B | -0.0001 | -0.0012 | -0.7% | -1.2% | +0.0% | +0.53% | -0.34% |
| OPT-6.7B | +0.0001 | -0.0006 | +0.4% | +0.2% | +0.0% | +0.0% | +0.15% |
| BLOOM-1.1B | -0.0001 | -0.0004 | +0.1% | +0.4% | +0.5% | / | +0.33% |
| BLOOM-1.7B | -0.0001 | -0.0002 | -0.16% | -0.8% | +0.5% | / | -0.15% |
| BLOOM-3B | -0.0002 | -0.0011 | +0.1% | -0.7% | +0.5% | / | -0.03% |

## 4.3 ABLATION STUDY

In this section, we aim to investigate the following questions: a) Which neurons are the most important for gender bias? FFN value neurons, attention value neurons, or FFN query neurons? b) Does the hyper-parameter M/N/P (number of edited neurons) affect the performance? c) What is the role of neuron filtering? d) Does editing method affect model performance? We conduct experiments on Llama-7B on the validation sets.

The results are shown in Table 6. The first line shows the results of the original model. Look at the results within the second, third and fourth blocks. When editing the FFN value neurons and the attention value neurons, the gender bias is reduced without hurting the common task performance. The reduction of gender bias achieves the best when directly editing the ffn query neurons without neuron filtering. However, the accuracy of other common tasks is also affected. For instance, when P=10, the RACE accuracy drops from 63.5 to 32.0 and the arithmetic score decreases from 51.86 to 7.43. Therefore, there are several "general neurons" not only important for gender bias but also for other tasks. When removing the general neurons in the identified query neurons, the common tasks are not affected. This proves the effectiveness of the neuron filtering stage of our Interpret-ME method, as our goal is mitigating the gender bias without affecting the model's abilities.

Comparing the overall results among the 2-4 blocks, the number of edited top neurons (M/N/P) can affect the performance. The results when M/N/P=5 are better than those when M/N/P=2. However, the improvement becomes less when increasing M/N/P to 10. When editing the top10 FFN value neurons, the StereoSet ICAT score drops from 61.40 to 60.32. This result indicates that the value neurons are more concentrated than query neurons. Hence, our final method is choosing M=5 and N=10 for FFN/attention value neurons and P=10 for FFN query neurons (the first line in the last block), performing the best score among all the results.

Lastly, we compare the zero-editing method with division-editing method in the last block. The performance of division-editing method changes when the division score is different. The ICAT score on SteroSet when D=10 decreases slightly compared with D=30. When increasing D to 30 and 100, all the results are similar compared with zero-editing. Therefore, we utilize the zero-editing method in all the other settings, as it does not require choosing the division score D.

Overall, from the results in Table 6 we can answer the questions in the begining of this section. a) FFN query neurons affect the gender bias most, but they also affect other common tasks. FFN value neurons and attention value neurons store gender bias parameters without hurting other tasks' performance. b) The hyper-parameter M/N/P can affect the performance and should be different for value neurons and query neurons. c) Neuron filtering is essential for selecting the gender neurons in query neurons, as there are also "general neurons" affecting other tasks' performance. d) Editing methods does not affect model performance much, and zero-editing is a good choice for starting.

Table 6: Ablation study of different settings. **Neu**: edited neurons (ori: the origin model; ffnv: editing FFN value neurons; attnv: editing attn value neurons; ffnq: editing FFN query neurons; all: ffnv & attnv & ffnq). **edit**: editing method (zero editing/division editing; D: division score). **MNP**: number of edited top neurons. **F**: whether the edited neurons are filtered. Metric: StereoSet: ICAT (larger better); WinoG/CPairs: entropy difference (smaller better); Others: accuracy (larger better).

| Neu | edit | MNP | F | Stero | WinoG | CPairs | PIQA | ARC | RACE | arithm |
|---|---|---|---|---|---|---|---|---|---|---|
| ori | - | - | - | 59.54 | 0.0095 | 0.0226 | 78.83 | 70.70 | 63.5 | 51.86 |
| ffnv | 0-edit | M=2 | ✗ | 60.47 | 0.0095 | 0.0228 | 78.78 | 70.70 | 63.5 | 51.85 |
| attnv | 0-edit | N=2 | ✗ | 59.54 | 0.0094 | 0.0224 | 78.67 | 70.70 | 63.5 | 51.91 |
| ffnq | 0-edit | P=2 | ✗ | 62.33 | 0.0068 | 0.0207 | 77.31 | 71.05 | 60.5 | 49.33 |
| ffnq | 0-edit | P=2 | ✓ | 59.54 | 0.0091 | 0.0225 | 78.78 | 70.70 | 63.5 | 51.86 |
| ffnv | 0-edit | M=5 | ✗ | 61.40 | 0.0095 | 0.0229 | 78.89 | 70.17 | 63.5 | 51.9 |
| attnv | 0-edit | N=5 | ✗ | 59.54 | 0.0094 | 0.0224 | 78.78 | 70.88 | 63.5 | 51.9 |
| ffnq | 0-edit | P=5 | ✗ | 63.26 | 0.0064 | 0.0204 | 77.31 | 71.22 | 60.5 | 49.36 |
| ffnq | 0-edit | P=5 | ✓ | 62.49 | 0.0087 | 0.0222 | 78.72 | 70.70 | 63.5 | 51.86 |
| ffnv | 0-edit | M=10 | ✗ | 60.32 | 0.0093 | 0.0228 | 78.94 | 70.52 | 64.0 | 51.9 |
| attnv | 0-edit | N=10 | ✗ | 61.24 | 0.0092 | 0.0226 | 78.78 | 70.87 | 63.5 | 51.61 |
| ffnq | 0-edit | P=10 | ✗ | 65.72 | 0.0102 | 0.0217 | 68.28 | 50.17 | 32.0 | 7.43 |
| ffnq | 0-edit | P=10 | ✓ | 61.56 | 0.0086 | **0.0220** | 78.73 | 70.70 | 63.5 | 51.9 |
| all | 0-edit | 5+10 | ✓ | **65.46** | **0.0084** | 0.0224 | 78.89 | 70.35 | 63.5 | 51.75 |
| all | D=10 | 5+10 | ✓ | 65.29 | 0.0084 | 0.0224 | 78.99 | 70.35 | 63.5 | 51.76 |
| all | D=30 | 5+10 | ✓ | 65.46 | 0.0084 | 0.0224 | 78.94 | 70.35 | 63.5 | 51.76 |
| all | D=100 | 5+10 | ✓ | 65.46 | 0.0084 | 0.0224 | 78.89 | 70.35 | 63.5 | 51.75 |

## 4.4 WHY DOES EDITING JUST A FEW NEURONS SIGNIFICANTLY REDUCE ACCURACY?

From the results in Section 4.3, it is surprising that editing only a few query neurons (when P=10) can result in a significant decrease in all the common tasks. In this section, we aim to explore the reason of this phenomenon. We find that the decreases in Llama-7B are mainly caused by two neurons in the 2nd FFN layer: $F_{4090}^2$ and $F_{7003}^2$. When editing these two neurons, the scores on PIQA, ARC, RACE and arithmetic are 68.17, 50.70, 31.5 and 7.51, respectively. The accuracy drops the most on arithmetic dataset, from 51.86 to 7.51.

To find the reason of this decrease, we use the comparable neuron analysis (CNA) method (Yu & Ananiadou, 2024) to analyze the change of the important neurons before and after the neurons $F_{4090}^2$ and $F_{7003}^2$ are edited. We analyze the case "3+5=" between the original model and the edited model. The prediction with the largest probability changes from "8" to "1". We compare the coefficient scores of the important neurons for the case "3+5=" identified by Yu & Ananiadou (2024), as shown

in Table 7. We find that the important neurons' coefficient scores are affected very much. For example, the coefficient score of $F_{5769}^{19}$ decreases from 3.79 to 0.48. Moreover, the sign of three neurons' coefficient scores are reversed, causing the probability from "increasing" to "decreasing". In comparison, when editing a gender neuron $F_{2026}^4$ (in Table 2), the important neurons' coefficient scores only changes 0.8% on average, thus the final prediction of "3+5=" is still "8".

Based on these observations, we conclude that the reason why the arithmetic accuracy drops much is that the important neurons' coefficient scores are changed (e.g. $F_{2258}^{11}$, $F_{4072}^{12}$, $F_{5769}^{19}$) by the edited general neurons ($F_{4090}^2$ and $F_{7003}^2$), because shallow neurons can affect deeper neurons. This analysis can also prove that the neuron filtering stage of our Interpret-ME method is essential.

Table 7: Change of important neurons' coefficient scores (**coeff**) in case "3+5=" **before/after** the general neurons ($F_{4090}^2$ and $F_{7003}^2$) are edited.

| neuron | coeff before/after | top tokens in unembedding space |
|--------|--------------------|---------------------------------|
| $F_{2258}^{11}$ | 0.09/-0.01 | [XV, fifth, Fif, avas, Five, five, abase, fif] |
| $F_{4072}^{12}$ | 0.04/-0.02 | [III, three, Three, 3, triple] |
| $F_{5769}^{19}$ | 3.79/0.48 | [eight, VIII, 8, III, huit, acht] |
| $F_{7164}^{25}$ | 8.43/3.97 | [six, eight, acht, Four, twelve, six, four, vier] |
| $F_{3696}^{28}$ | 6.20/-0.01 | [8, eight, VIII, huit, acht, otto] |

## 5 DISCUSSION: WHY ARE NEURON CIRCUITS IMPORTANT?

In this section, we aim to discuss the importance of constructing the neuron circuits.

**a) More precise parameter localization can help preserve a model's existing capabilities during model editing.** Each layer in a LLM contains thousands of neurons, with different neurons potentially contributing to various tasks or word representations. As we discuss in Section 4.4, even editing just two neurons can significantly impact the model's abilities, as changes in shallow neurons can propagate and influence deeper ones. Therefore, model editing should be approached with caution. Our Interpret-ME method offers a neuron-level editing approach. When we observe a performance decline in the edited model, we can analyze the neurons individually to identify which ones are critical for other tasks and restore those neurons to mitigate the impact.

**b) Neuron circuits provide deeper insights into underlying mechanisms.** Due to the phenomenon of superposition (Elhage et al., 2022), directly analyzing individual neurons makes it challenging to determine their specific roles. Most interpretability methods involve projecting neurons into unembedding space, where each neuron is associated with certain "top tokens". However, this can be misleading, as not all neurons directly contribute to the final predictions. If we project query neurons into unembedding space without the transformation by attention heads, the resulting top tokens may not accurately reflect the neurons' actual function. Additionally, when encountering a neuron, it is unclear whether it functions as a query neuron or a value neuron, nor is it evident which attention head is performing the transformation. Therefore, a more effective approach is to identify neuron circuits across diverse contexts and assess the significance of neurons in specific tasks or multiple scenarios. In this situation, the functions of the identified neurons are clear. By examining these circuits across various sentences, we can more precisely determine the function of each neuron.

## 6 CONCLUSION

Although LLMs gain powerful abilities from large amounts of data, they can also learn, perpetuate, and amplify biases. In this paper, we propose the Interpretable Model Editing (Interpret-ME) method to mitigate gender bias in LLMs without designing new datasets or fine-tuning. Based on interpretability analysis of gender-biased sentences, we find that several neurons contain much gender bias. Our Interpret-ME method has three stages: neuron locating, neuron analyzing and filtering, and neuron editing. We conduct experiments on 8 LLMs using three gender bias datasets, and our method shows competitive performance compared to fine-tuning methods. Additionally, we perform experiments on four general tasks and find that our method does not compromise their performance. Overall, our method and analysis are crucial for understanding the mechanism of gender bias and offer a potential solution for mitigating the gender bias.

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

## A  APPENDIX A: GENDER NEURONS IN OPT AND BLOOM

We also use the sentence "The nurse is a => woman" to locate the gender neurons in OPT and BLOOM. The gender neurons in OPT and BLOOM are shown in Table 8.

Table 8: Identified gender neurons in OPT (first block) and BLOOM (second block).

| neuron | model | top tokens | last tokens |
|--------|-------|-----------|-------------|
| $F_{6674}^{27}$ | OPT | [wife, Wife, spokeswoman, she, wives] | [gentlemen, brothers, father, guy, boys, brother] |
| $A_{45}^{27,1}$ | OPT | [she, her, herself, she, hers, woman, She, daughter, Women] | [Mr, himself, his, Adam, Michael, Jason, frontman, Mike] |
| $F_{5484}^{7}$ | OPT | [girl, Girl, girls, Girls, she, her, girl, feminist, woman, herself] | [son, Mr, fathers, his, grandson, dads, sons, Mr, dad, father] |
| $F_{8640}^{23}$ | BLOOM | [woman, women, lady, girl] | [masculina, mascul, masculino, himself, male, masculine] |
| $F_{1407}^{27}$ | BLOOM | [lady, woman, femme, women, mujer, girl, femmes, women's] | [Hombre, Policia, father, man] |

## B  APPENDIX B: GENDER BIAS SENTENCES FOR IDENTIFYING GENDER NEURONS

To identify the important neurons containing gender bias, we only use five sentences for each gender. ALL the sentences follow the pattern "The XX is a", where "XX" is a profession, shown in Table 9.

Table 9: Gender bias sentences for identifying important neurons.

| Male | Female |
|------|--------|
| police, guard, delivery, driver, machinist | nurse, domestic helper, seller, librarian, beautician |

