# OpenReview forum: "Understanding and Mitigating Gender Bias in LLMs via Interpretable Model Editing"
_ICLR.cc/2025/Conference — Submitted to ICLR 2025_

### Official Review · Reviewer_wmNw · 2024-10-25

**Soundness:** 3
**Presentation:** 4
**Contribution:** 3
**Rating:** 8
**Confidence:** 3

**Summary:**

This paper introduces the Interpretable Model Editing (Interpret-ME) method, which effectively reduces gender bias in LLMs by identifying key neurons without requiring large datasets or fine-tuning, achieving competitive results while preserving performance in other tasks.

**Strengths:**

1.	This paper analyzes the neurons in LLMs responsible for storing gender bias, contributing to a deeper understanding of how gender bias exists within these models.
2.	By identifying key neurons associated with gender bias, the paper demonstrates that editing these neurons can achieve better debiasing results with minimal impact on overall model performance.
3.	The paper also explores the importance of different neurons and points out that "FFN query neurons" have the most significant influence on gender bias.

**Weaknesses:**

The located neurons may not be sufficiently representative. Since only five sentences per gender are used to locate neurons, these sentences may not adequately capture real-world gender stereotypes. Moreover, there is no experiment in the paper that demonstrates whether using more or fewer sentences would affect the performance of the Interpret-ME method.

**Questions:**

Why is Table 5 not a comparison between the Interpret-ME method, fine-tuning methods, and the original model? Why does Table 5 not include a comparison between the Interpret-ME method, fine-tuning methods, and the original model? I would like to know whether Interpret-ME causes less performance drop compared to fine-tuning methods.

---

> ### Author Response · Authors · 2024-11-15
> **response to reviewer wmNw**
>
> Thank you very much for your valuable comments. We thank you for understanding the strengths of our work regarding the analysis of neuron-level information flow and the neuron-level model editing.
>
> Regarding the weaknesses, here are our responses:
>
> _Q: The located neurons may not be sufficiently representative. Since only five sentences per gender are used to locate neurons, these sentences may not adequately capture real-world gender stereotypes. Moreover, there is no experiment in the paper that demonstrates whether using more or fewer sentences would affect the performance of the Interpret-ME method._
>
> A: We agree that the located neurons may not be sufficiently representative. However, we think that **this is one of the advantages of our work: we do not need to analyze all the probable sentences and identify all the important neurons. Only editing the neurons in 10 sentences can reduce the gender bias very much.** All the gender bias datasets in Section 4 are general gender bias datasets, rather than designed datasets. Different gender bias cases are contained in these datasets. The experimental results in Ssection 4 can prove that the identified neurons are important for these datasets and these different sentences. When zero-editing the neurons, the gender bias is reduced much.
>
> To boost the understanding about the mechanism of gender bias, we analyze the cases "The strong person is a" -> "man", "The beautiful person is a" -> "woman". We find that **most of the activated neurons are the same with the important neurons identified by the professions**. For instance, Table 1's attention neuron's coefficient score is 0.5281 ($A^{18, 7}_{54}$) in "The beautiful person is a" -> "woman".
>
> This is not surprising because the interpretability results can prove that this neuron contains the information about increasing "woman" probability and decreasing "man" probability when the coefficient score is larger than zero. Furthermore, we analyze the case "The woman is a" -> "nurse". In this case, the shallow FFN neurons and the attention neurons are similar. For example, **the shallow FFN neuron $F_{2026}^4$ is activated (coefficient: 0.0108) by the word "woman", and added into this position's residual stream. Then the attention neuron $A^{18, 7}_{83}$ (coefficient: 0.0361) is activated by the woman position's hidden states.** The deep FFN neurons are different, because these identified neurons contain the information about the word "nurse", rather than "woman". These results indicate that different cases can activate similar neurons.
>
> Regarding the number of sentences, we will add the experiments about the number of the sentences. We conducted the experiments on Llama-7B with number 1,2,5,10,20,30,40 for each gender. The metrics increase sharply from 1 to 5, increase slowly from 5 to 20, and starts to decrease from 20.
>
> _Q: Why is Table 5 not a comparison between the Interpret-ME method, fine-tuning methods, and the original model? Why does Table 5 not include a comparison between the Interpret-ME method, fine-tuning methods, and the original model? I would like to know whether Interpret-ME causes less performance drop compared to fine-tuning methods._
>
> A: We cannot get the model parameters of the fine-tuned models in [1]. Therefore, we cannot conduct the experiments to compare the results.
>
> [1]  A trip towards fairness: Bias and de-biasing in large language models

---

> > ### Comment · Reviewer_wmNw · 2024-11-27
> >
> > Thank you for your response. Most of my concerns have been addressed, so I plan to maintain my current rating. I'm interested in the ongoing conversation with Reviewer W11j as well, I hope that will be resolved before the closed discussion.

---

> > > ### Author Response · Authors · 2024-11-27
> > >
> > > Thank you very much for your supportive responses. We also aim to address Reviewer W11j's concerns thoroughly. To this end, we have conducted additional experiments and analysis, and we believe the results provide clear and substantial insights.

---

### Official Review · Reviewer_W11j · 2024-11-02

**Soundness:** 2
**Presentation:** 3
**Contribution:** 2
**Rating:** 3
**Confidence:** 4

**Summary:**

This paper mitigates the gender bias issue of large language models by editing model parameters instead of data cleaning and fine-tuning. The paper argues that some neurons in LLM exhibits significant bias and thus results in the bias of LLM. Therefore, the authors firstly adopt interpretability methods to identify such neurons and then propose Interpret-ME to reduce it. The experimental results demonstrate the effectiveness of Interpret-ME across 8 LLMs without degrading their performance.

**Strengths:**

1. It is important to studying the neuron circults of the generated response to make LLMs more interpretable and align with human values. The motivation is convincing.
2. The methodology of implementing the idea is sensible.
3. The experimental results show the effectiveness of the methods.

**Weaknesses:**

1. My major concern is that the models studied by the authors may be outdated. With the continuous development of LLMs, more and more drawbacks of them vanished. It’s unclear whether recent LLMs suffer from such an issue and whether the proposed method could generalize to recent LLMs. I suggest more experiments to clarify this point. Otherwise, the contribution of this work will be vague or limited.
2. It seems that the activated neurons vary across different prompts, and gender bias is often implicitly represented by models. Is the women-and-man (or specific-prompt) setting generalizable and convincing enough to arrive at the research conclusions? Are the adopted settings representative enough?

**Questions:**

1. Do recent LLMs suffer from gender bias issues (e.g., o1, GPT-4o, GPT-4, GPT-3.5-turbo, LLaMa 3.1, LLaMa 3.2)?
2. With the recent development of LLMs, their “intelligence” keeps growing due to the boost of data volume and quality. Various kinds of bias are less likely to be present in the training data. Could you give some examples of the bias categories represented by recently proposed LLMs?

---

> ### Author Response · Authors · 2024-11-15
> **response to reviewer W11j**
>
> Thank you very much for your valuable comments. And thank you for understanding the strengths about our work, regarding the importance of the task, the correctness and effectiveness of our proposed method.
>
> _Q: My major concern is that the models studied by the authors may be outdated. With the continuous development of LLMs, more and more drawbacks of them vanished. It’s unclear whether recent LLMs suffer from such an issue and whether the proposed method could generalize to recent LLMs. I suggest more experiments to clarify this point. Otherwise, the contribution of this work will be vague or limited._
>
> A: We understand that your main concern is: is this work still useful? Here are our responses:
>
> Firstly, **gender bias is obtained by LLMs during pre-training.** [1] find that the co-occurrence of word pairs is important for LLMs to predict the answers. [2] find that LLMs learn the associate certain professions with specific genders based on gender stereotypes, but these data’s distributions are not “wrong” for the model. For example, if the co-occurrence of “man” and “guard” is larger than that of “woman” and “guard”, the output of “The guard is a” is more likely to be “man” than “woman”. Under this mechanism, **it is very hard to reduce the gender bias during pre-training**. Experimentally, [3] find that the gender bias is the hardest to be reduced compared with other bias (e.g. racism).
>
> Secondly, most current methods try to reduce the gender bias during SFT or RLHF. However, [4] find that the **capabilities learned from pre-training are not removed, but rather bypassed. In other words, the parameters storing toxicity/bias still exist in the fine-tuned model. When LLMs get unseen inputs or designed prompts, they will still generate toxicity/bias outputs.**
>
> Based on the mechanism analysis, gender bias cannot be removed during pre-training or SFT/RLHF. This is the reason why we use model editing method to solve this problem. **The gender bias parameters are stored in neurons, and our work can reduce the gender bias by zero-editing these neurons.**
>
> Besides, we do the same experiments with Table 4 on Llama 3.1-8B. The lms/ss/nss/icat scores of Llama 3.1 are 94.54/68.81/62.37/58.97. **The performance is even worse than Llama-7B**. This means that Llama-3.1-8B still has gender bias, and it is not better than Llama-7B. This conclusion is similar to that of [3]: **gender bias is not reduced in LLMs with better abilities.**
>
> **Can our method be utilized in Llama-3.1-8B? The answer is yes**. We apply our method on Llama-3.1-8B and the lms/ss/nss/icat scores are 94.63/67.93/64.13/60.69, which is better than that of the original model.
>
> _Q: It seems that the activated neurons vary across different prompts, and gender bias is often implicitly represented by models. Is the women-and-man (or specific-prompt) setting generalizable and convincing enough to arrive at the research conclusions? Are the adopted settings representative enough?_
>
> A: Actually, **the activated neurons are similar under different prompts**. The results of Table 1, 2, and 3 can prove that the sentences with different genders can activate the same neurons. Only the coefficient scores of the neurons are different. When we zero-edit these neurons, the gender bias is reduced. Furthermore, all the datasets in Section 4 are general gender bias datasets contain different prompts and cases (e.g. personality traits). The experimental results can also prove that our method is not only useful for the designed prompts, but also useful for different gender bias sentences.
>
> _Q: Do recent LLMs suffer from gender bias issues (e.g., o1, GPT-4o, GPT-4, GPT-3.5-turbo, LLaMa 3.1, LLaMa 3.2)?_
>
> A: First, it is important to note that our work is a mechanistic interpretability work, which needs to locate and edit the important neurons. **Compared with previous works which regard LLMs as blackboxes, it is much harder to analyze the inner mechanism in LLMs and leverage the inpretability findings to solve real tasks**. In previous mechanistic interpretability works, the experiments are usually conducted on small models such as GPT2 [5,6]. It is a breakthrough to do the neuron-level model editing in LLMs.
>
> Because our work needs to locate and edit the inner parameters, we cannot do experiments on the close-source LLMs. But our experiments on Llama-3.1-8B can prove that the gender bias is still not reduced, and our work is useful for reducing gender bias.
>
> [1] Impact of Co-occurrence on Factual Knowledge of Large Language Models
>
> [2] Gender Bias and Stereotypes in Large Language Models
>
> [3] A trip towards fairness: Bias and de-biasing in large language models
>
> [4] A Mechanistic Understanding of Alignment Algorithms: A Case Study on DPO and Toxicity
>
> [5] Locating and editing factual associations in GPT
>
> [6] How does GPT-2 compute greater-than?: Interpreting mathematical abilities in a pre-trained language model

---

> ### Comment · Reviewer_W11j · 2024-11-27
>
> Thanks for PC's feedback and the authors' response. The research is indeed very important, but my concern still has not been addressed at present.
>
> Regarding my major concern, I suppose that with the increase of both the diversity of training data and the number of parameters, gender bias will be significantly mitigated. If gender bias could be addressed through the scaling law, the value of this work would be limited. It's better to demonstrate that these models still suffer from gender bias to imply the promising potential of the proposed method.
>
> Regarding my second concern, researchers in the knowledge-editing domain demonstrate that the activated neurons vary across different prompts. What's the assumption here to conclude that the activated neurons are similar under different prompts? Do the prompts share some common regularities? In real-world scenarios, for example, will the 50 prompts (10 words per prompt) activate similar neurons when adding 50 different contexts (1000 words per context) before them, respectively? If the activated neurons change accordingly, how to use the model editing method to fix the gender bias issue in real-world scenarios?
>
> Even after the rebuttal, the issues were not completely resolved, so I will maintain my current rating.

---

> ### Author Response · Authors · 2024-11-27
>
> Thanks for your responses. We hope that the following responses can address your concerns.
>
> Q1: _Regarding my major concern, I suppose that with the increase of both the diversity of training data and the number of parameters, gender bias will be significantly mitigated. If gender bias could be addressed through the scaling law, the value of this work would be limited. It's better to demonstrate that these models still suffer from gender bias to imply the promising potential of the proposed method._
>
> A1: In our last response, we conducted the experiments in Llama3.1 on StereoSet dataset, similar to Table 4. **The lms/ss/nss/icat scores of Llama 3.1 are 94.54/68.81/62.37/58.97, which is similar to the results of other models in Table 4.** This can prove that gender bias can't be addressed through the scaling law. And our interpret-ME method is also useful to reduce the gender bias in Llama 3.1, the scores of interpret-ME are 94.63/67.93/64.13/60.69.
>
> Furthermore, the analysis in previous studies have similar results, and here are the analysis:
>
> Firstly, **gender bias is obtained by LLMs during pre-training.** [1] find that the co-occurrence of word pairs is important for LLMs to predict the answers. [2] find that LLMs learn the associate certain professions with specific genders based on gender stereotypes, but these data’s distributions are not “wrong” for the model. For example, if the co-occurrence of “man” and “guard” is larger than that of “woman” and “guard”, the output of “The guard is a” is more likely to be “man” than “woman”. **Under this mechanism, it is very hard to reduce the gender bias during pre-training. Experimentally, [3] find that the gender bias is the hardest to be reduced compared with other bias (e.g. racism).**
>
> Secondly, most current methods try to reduce the gender bias during SFT or RLHF. However, [4] find that **the capabilities learned from pre-training are not removed, but rather bypassed.** In other words, the parameters storing toxicity/bias still exist in the fine-tuned model. When LLMs get unseen inputs or designed prompts, they will still generate toxicity/bias outputs.
>
> Q2: _Regarding my second concern, researchers in the knowledge-editing domain demonstrate that the activated neurons vary across different prompts. What's the assumption here to conclude that the activated neurons are similar under different prompts? Do the prompts share some common regularities? In real-world scenarios, for example, will the 50 prompts (10 words per prompt) activate similar neurons when adding 50 different contexts (1000 words per context) before them, respectively? If the activated neurons change accordingly, how to use the model editing method to fix the gender bias issue in real-world scenarios?_
>
> A2: Previous studies' [5] conclusion is: **different knowledge is stored in different parameters, and similar knowledge is stored in similar parameters.** Their experiments were done in real datasets, rather than designed prompts. And this is the assumption of our work.
>
> When adding different prompts before the context, the activated neurons will be different because the prompts are different. However, in our work, **the important neurons are identified by both the inputs and the final predictions. In equation 8 and 9, the identified neurons contain the logits of the final predictions.** The identified neurons are the most important neurons affecting the probability of the final predictions. In other words, not all the activated neurons are selected for model editing, only the most important neurons affecting final predictions are identified. If the prompts do not contain gender bias, the neurons activated by the prompts will not be identified and selected.
>
> Following your suggestion, we add 10 different contexts before the original 10 gender sentences in our original paper, and analyze the difference of the identified top100 neurons. We find that **84% identified neurons are the same.** This result can also prove our assumption.
>
> Again, we hope to mention that **the experimental results in Table 4 and 5 can prove the correctness of our method. These datasets are real datasets rather than designed prompts. In these datasets, the sentences are very different, but the gender bias is reduced using our method.** This can prove that our method can identify the important neurons containing gender bias.
>
> [1] Impact of Co-occurrence on Factual Knowledge of Large Language Models, 2023
>
> [2] Gender Bias and Stereotypes in Large Language Models, 2023
>
> [3] A trip towards fairness: Bias and de-biasing in large language models, 2023
>
> [4] A Mechanistic Understanding of Alignment Algorithms: A Case Study on DPO and Toxicity, 2024
>
> [5] Neuron-Level Knowledge Attribution in Large Language Models, 2024

---

### Official Review · Reviewer_iAJp · 2024-11-03

**Soundness:** 3
**Presentation:** 2
**Contribution:** 2
**Rating:** 3
**Confidence:** 3

**Summary:**

This paper introduces an approach to mitigate gender bias using a neuron-level framework called Interpret-ME. Unlike resource-intensive finetuning methods, this proposed approach edits key neurons to reduce bias while maintaining model performance. The authors demonstrate the effectiveness of their proposed approach on eight LLMs using various metrics such as StereoSet, WinoGender, and CrowS-Pairs, achieving competitive results.

**Strengths:**

- The study addresses a crucial issue in machine learning: mitigating gender bias in LLMs through an efficient method that avoids resource-heavy fine-tuning or data collection.
- It is validated across a range of large and common models, enhancing practical significance.
- The method also maintains overall model performance while offering valuable neuron-level interpretability insights into bias mechanisms.

**Weaknesses:**

The paper requires substantial revisions to meet the standards of a prestigious venue like ICLR. The main issues are:

- The writing is difficult to follow due to unclear explanations and a confusing structure. Key sections, such as the background and methodology, require multiple readings to understand. For example, the background section introduces numerous variables and equations without sufficient context or motivation, making it challenging for readers to relate them to the main methodology. While it extensively reiterates standard multi-head attention formulas with many variables, it fails to explain their relevance to this work. Conversely, the authors omit essential background on the key concept of the unembedding space, which is central to understanding the proposed methodology.

- Figures and tables are presented without adequate spacing, blending into the text and making it difficult to differentiate between the main content and captions (e.g., page 5).

- The method relies heavily on existing interpretability frameworks, leading to limited innovation and making the contributions feel incremental.

- The idea of addressing bias in LLMs by identifying and editing specific neurons is not new, and similar approaches have been explored before. The authors do not adequately acknowledge this and fail to distinguish their work from related studies like those by Chintam et al. (2023) and Lutz et al. (2024). For example, Chintam et al. (2023) used methods like automated circuit discovery to identifying causal relations between LM components and gender bias, following by performing a finetuning strategy to mitigate bias in those components.

- The study's use of bias-evaluation datasets, such as CrowS-Pairs and StereoSet, which have been criticized for noise and reliability issues (Blodgett et al., 2021), raises concerns about the robustness of the evaluation.

[1] Chintam, A., Beloch, R., Zuidema, W., Hanna, M., & Van Der Wal, O. (2023). Identifying and adapting transformer-components responsible for gender bias in an English language model. arXiv preprint arXiv:2310.12611.

[2] Lutz, M., Choenni, R., Strohmaier, M., & Lauscher, A. (2024). Local Contrastive Editing of Gender Stereotypes. arXiv preprint arXiv:2310.17739.

[3] Blodgett, S. L., Lopez, G., Olteanu, A., Sim, R., & Wallach, H. (2021). Stereotyping Norwegian Salmon: An Inventory of Pitfalls in Fairness Benchmark Datasets. In Proceedings of the 59th Annual Meeting of the Association for Computational Linguistics and the 11th International Joint Conference on Natural Language Processing (Volume 1: Long Papers), pages 1004–1015, Online. Association for Computational Linguistics.

**Questions:**

Q1. Could you provide a clearer explanation of how the hypotheses in Section 3.3 were derived from the previous analyses?

---

> ### Author Response · Authors · 2024-11-15
> **response to reviewer iAJp**
>
> Thank you very much for your valuable comments. We thank you for understanding our work’s strengths regarding crucial issue, good experiments and good interpretability.
>
> Here are our responses regarding the weaknesses.
>
> _Q: The writing is difficult to follow ... the authors omit essential background on the key concept of the unembedding space, which is central to understanding the proposed methodology._
>
> A: Our work is a mechanistic interperetability work exploring the mechanism of gender bias. **In the field of mechanistic interpretability, projecting the internal hidden states and neurons in unembedding space is a commonly-used method** [1,2,3,4,5] to analyze the interpretability of different parameters. In some recent mechanistic interpertability works [6], these backgrounds are considered as common sense and even not introduced any more. In comparison, we introduce it in lines 188-193, and Section 2.2. The audience can understand the background well from these lines.
>
> _Q: Figures and tables ... (e.g., page 5)._
>
> A: We will modify the format.
>
> _Q: The method relies heavily on existing interpretability frameworks, leading to limited innovation and making the contributions feel incremental._
>
> A: **The understanding about how LLMs work will not happen overnight.** Therefore, using the existing interpretability frameworks is essential. However, we do not agree that our work has limited innovation and incremental contributions. First, regarding the mechanistic interpretability of gender bias, **the neuron-level information flow about gender bias is not explored.** Most works aim to evaluate the gender bias by reducing the metrics on gender bias datasets, but the mechanism is not explored much. In other words, **the neuron-level information flow of gender bias is still not clear. Our study provides a deep understanding about gender bias.**  Second, we propose a model editing method to **use the interpretability finds to solve real problem**, which is a breakthrough compared with most interpretability works.
>
> _Q: The idea of addressing bias in LLMs by identifying and editing specific neurons is not new, and similar approaches have been explored before. The authors do not adequately acknowledge this and fail to distinguish their work from related studies like those by Chintam et al. (2023) and Lutz et al. (2024)..._
>
> A: We will introduce the works in Section 2. Our work aims to understand and reduce the gender bias in LLMs. But **these works cannot be applied in LLMs, which do not fit our work's situations.** The methods in the mentioned works can only be applied in small models like GPT-2 and BERT due to the **computational costs**. They apply the causal-based methods and circuit discovery methods to identify the important parameters. However, these methods cannot be applied at neuron-level in large language models, because **there are too many neurons in LLMs.** Take Llama-7B as an example, there are 483,328 neurons in the model. If using causal-based methods, the forward computation stage for identifying the important neurosn should be done 483,328 times.
>
> _Q: The study's use of bias-evaluation datasets, such as CrowS-Pairs and StereoSet, which have been criticized for noise and reliability issues (Blodgett et al., 2021), raises concerns about the robustness of the evaluation._
>
> A: **We choose the same datasets with the most popular LLMs.** These datasets are commonly used to evaluate LLMs. For example, CrowS-Pairs is used in Llama and GLM. And StereoSet is used in GPT3 and GLM.
>
> _Q: Could you provide a clearer explanation of how the hypotheses in Section 3.3 were derived from the previous analyses?_
>
> A: In FFN layers and attention heads, the subvalues (fc2 in Eq.5) are the same in all the cases. The changing thing is the coefficient score of each neuron (m in Eq.5). In Table 1 and Table 2, **if identified neurons’ top tokens are related to “man”, their last tokens are related to “woman”. If these neurons’ top tokens are related to “woman”, their last tokens are related to “man”.** Therefore, the neurons store the gender bias. **When the coefficient scores are larger than zero, the top tokens’ probability increase and the last tokens’ probability decrease. When the coefficient scores are smaller than zero, the top tokens’ probability decrease and the last tokens’ probability increase.** Furthermore, in Table 3 we can see that the sign of the coefficient scores on the same neuron is different under different genders, which matches our analysis.
>
> [1] interpreting GPT: the logit lens, 2020
>
> [2] Transformer Feed-Forward Layers Build Predictions by Promoting Concepts in the Vocabulary Space,  2022
>
> [3] Analyzing Transformers in Embedding Space, 2022
>
> [4] Future Lens: Anticipating Subsequent Tokens from a Single Hidden State, 2023
>
> [5] Neuron-Level Knowledge Attribution in Large Language Models, 2024
>
> [6] Dissecting Recall of Factual Associations in Auto-Regressive Language Models, 2023

---

> > ### Comment · Reviewer_iAJp · 2024-11-23
> >
> > - The term "unembedding space" is not a standard term in most NLP literature, and even in the references you provided, it does not appear explicitly. While I understand your argument, I believe there is significant room for improvement in presenting the flow of information more coherently to make your methodology clearer and more accessible to readers.
> >
> > - Regarding the related works I mentioned, I strongly recommend that you acknowledge and compare your work to this existing body of research. I find your explanation for not including these works unconvincing, especially since they share notable similarities with your approach. Like reviewers rUmF and wmNw, I share concerns about the validity of your results. A detailed comparison with previous studies is essential to validate your method and demonstrate its distinctiveness.
> >
> > - The bias metrics you used have been criticized for reliability issues. I suggest you to expand the range of evaluation metrics to ensure the robustness of your findings.
> >
> > Given the importance of these issues, I must regretfully maintain my initial scores.

---

> > > ### Author Response · Authors · 2024-11-23
> > >
> > > Thanks for your reply.
> > >
> > > _Q: The term "unembedding space" is not a standard term in most NLP literature, and even in the references you provided, it does not appear explicitly. While I understand your argument, I believe there is significant room for improvement in presenting the flow of information more coherently to make your methodology clearer and more accessible to readers._
> > >
> > > A: "Unembedding space" is a key concept in mechanistic interpretability. We recognize that audiences unfamiliar with this field might find it challenging to understand, so we have provided references for further reading. If the concept remains unclear, we trust that these resources will help clarify it.
> > >
> > > In essence, "unembedding space" refers to the vector space defined by the unembedding matrix. The next token's distribution is computed by multiplying the final layer's output at the last position with this matrix. Projecting a vector into unembedding space involves multiplying the vector by the unembedding matrix and analyzing the resulting token rankings. A top-ranked token indicates that the vector increases the token's probability, while a low-ranked token suggests the opposite. This concept originates from the 2020 "logit lens" work, as cited in our references.
> > >
> > > _Q: Regarding the related works I mentioned, I strongly recommend that you acknowledge and compare your work to this existing body of research. I find your explanation for not including these works unconvincing, especially since they share notable similarities with your approach. Like reviewers rUmF and wmNw, I share concerns about the validity of your results. A detailed comparison with previous studies is essential to validate your method and demonstrate its distinctiveness._
> > >
> > > A: We have taken your advice into account. In Section 2, we will introduce these related works and provide a detailed discussion of the differences between their approaches and ours. Notably, we will explain why these methods cannot be directly applied to LLMs due to their computational costs, which highlights the unique contributions of our study.
> > >
> > > _Q: The bias metrics you used have been criticized for reliability issues. I suggest you to expand the range of evaluation metrics to ensure the robustness of your findings._
> > >
> > > A: Although these datasets and metrics have been criticized, they are still widely used in recent studies (e.g., the two works you mentioned). Therefore, conducting experiments on them is essential to ensure comparability with other research.

---

### Official Review · Reviewer_rUmF · 2024-11-03

**Soundness:** 2
**Presentation:** 3
**Contribution:** 3
**Rating:** 5
**Confidence:** 3

**Summary:**

Existing approaches usually use model re-training or model fine-tuning methods to alleviate gender bias. They usually require curating a data set for debiasing purposes. And such re-training and fine-tuning might hurt model’s performance on other tasks.
Towards this end, the paper proposes Interpretable Model Editing (Interpret-ME), a method to reduce gender bias without designing huge datasets or fine-tuning. Compared to fine-tuning methods, the proposed approach shows competitive results in reducing gender bias across experiments with 8 LLMs.

**Strengths:**

1. good presentation

2. proper adaptation of existing methods to application problems

3. thorough experiments on various models

**Weaknesses:**

My only concern is that from the experiments, it seems that in order for Interpret-ME to not hurt models’ performance on other tasks, it requires very delicate hyper-parameter search. Therefore I’m not convinced that compared to fine-tuning approaches, the proposed approach is more beneficial from the perspective of maintaining LLM’s existing capability. More experiment designs and results along this line would be very helpful.

**Questions:**

Please see weakness.

---

> ### Author Response · Authors · 2024-11-15
> **response to reviewer rUmF**
>
> Thank you very much for your valuable feedbacks. We thank you for understanding our work’s strengths about good representation, proper method, and thorough experiments.
>
> Here are our responses regarding the weeknesses.
>
> _Q: My only concern is that from the experiments, it seems that in order for Interpret-ME to not hurt models’ performance on other tasks, it requires very delicate hyper-parameter search. Therefore I’m not convinced that compared to fine-tuning approaches, the proposed approach is more beneficial from the perspective of maintaining LLM’s existing capability. More experiment designs and results along this line would be very helpful._
>
> A: Firstly, **our method does not require the “very delicate hyper-parameter search”.** The interpretability analysis in Section 3.2 and the experiments in Section 4.3 is an ablation study to understand the roles about different neurons. **In model editing stage, the neuron selection is done automatically by calculating the neurons’ top tokens in unembedding space**, which are introduced in Section 3.3, lines 278-279.
>
> Secondly, **our method has the following advantages compared with fine-tuning.** a) Our method achieves **better performance** in Table 4 and Table 5. b) Our method **does not require much data**. We only needs 10 cases to identify the neurons, and the results are good on all the 8 models. c) Our method is **much faster**. The neuron selection stage only takes 20-30 seconds for one case.
>
> Thirdly, **previous methods have proved that fine-tuning cannot solve the gender bias problem**. As we introduced in lines 38-51 in Section 1, gender bias is not reduced much during fine-tuning [1]. [2] investigates that the fine-tuning data for reducing gender bias can bring factuality errors and potential risks. [3] point out that in-training methods risk corrupting the pre-trained language understanding due to catastrophic forgetting. Based on these problems, using other methods rather than fine-tuning is essential. In this work, we design the interpretable model editing method, and the experimental results are good.
>
> Fourthly, previous interpretability research [4] explore the mechanism of toxicity and find that **capabilities learned from pre-training are not removed, but rather bypassed**. In other words, **the parameters storing the toxicity still exist in the fine-tuned model. Undering prompts that are unseen in the fine-tuning data, these parameters can still be activated and causing the toxicity sentences.** The situation in gender bias is similar. **Our interpretable model editing method is a good way to solve this problem. We locate the important paramters and delete them.** Since different gender bias cases activate similar gender bias neurons, the gender bias will not be activated by unseen sentences. And this is why our method can achieve good results when we only use 10 gender bias cases to locate the neurons.
>
> In conclusion, according to previous studies, fine-tuning is not enough for reducing gender bias. Our interpretability analysis explores the mechanism of gender bias, and reduce the gender bias by editing the important neurons. Compared with fine-tuning methods, our method is faster because the neruon-selected stage of our method is automatic, and our method requires much less data than fine-tuning.
>
> [1] A trip towards fairness: Bias and de-biasing in large language models
>
> [2] Language generation models can cause harm: So what can we do about it? an actionable survey
>
> [3] Bias and fairness in large language models: A survey
>
> [4] A Mechanistic Understanding of Alignment Algorithms: A Case Study on DPO and Toxicity

---

### Official Review · Reviewer_jVvc · 2024-11-04

**Soundness:** 2
**Presentation:** 3
**Contribution:** 3
**Rating:** 6
**Confidence:** 4

**Summary:**

This paper explores gender bias in large language models (LLMs) and highlights challenges with current methods, like data cleaning and fine-tuning, which become costly as models get larger. The authors use interpretability tools to identify specific neurons linked to gender bias and introduce a new method, Interpretable Model Editing (Interpret-ME), to reduce this bias without needing extensive retraining.

**Strengths:**

1. The method does not rely on a large amount of data, making it more practical and cost-effective compared to approaches that require extensive datasets for bias mitigation.

2. Since the method does not require fine-tuning the entire model, it saves substantial computational resources and time, especially for large language models.

3. The method has minimal impact on performance across common datasets, ensuring that the model’s general abilities remain intact while reducing gender bias.

**Weaknesses:**

1. Some notations can be more clear. For example, B and d in section 3.1.

2. The method does not compare changes in entropy difference on WinoG/CPairs with fine-tuning. Without this comparison, it is unclear if Interpret-ME is as effective as or better than fine-tuning in terms of reducing bias on these datasets.

3. It remains unclear whether different gender-biased sentences activate the same neurons or if varying sentences affect the method's results. This uncertainty suggests that the method might not generalize well to a broad range of gender-biased language, potentially impacting its consistency and reliability across diverse examples.

**Questions:**

1. Will different types of gender-biased sentences activate distinct important neurons? The selected sentences focus on professions. If sentences featuring other gender-stereotyped topics, such as personality traits or colors, are used, would we observe similar results?

2. What happens if the sentence is changed to "This woman is ==> a nurse"?

---

> ### Author Response · Authors · 2024-11-15
> **response to Reviewer jVvc**
>
> Thank you very much for your valuable feedback. First, we appreciate your recognition of the strengths of our work. Our method requires minimal data and operates at a fast speed. Moreover, it preserves the model's original capabilities while effectively reducing gender bias. Regarding the weaknesses, here are our responses:
>
> _Q: Some notations can be more clear. For example, B and d in section 3.1._
>
> A: In Section 3.1, we introduce in lines 152-154 that B is the number of the tokens in unembedding space. D is the dimension of each word embedding. We follow the introduction of recent publications [1,2].
>
> _Q: The method does not compare changes in entropy difference on WinoG/CPairs with fine-tuning. Without this comparison, it is unclear if Interpret-ME is as effective as or better than fine-tuning in terms of reducing bias on these datasets._
>
> A: The results of the fine-tuning method comes from [3], the state-of-the-art fine-tuning method for reducing gender bias. They only do the experiments on StereoSet, and we cannot get the parameters of the fine-tuned models to compare the other experiments with them.
>
> _Q: It remains unclear whether different gender-biased sentences activate the same neurons or if varying sentences affect the method's results. This uncertainty suggests that the method might not generalize well to a broad range of gender-biased language, potentially impacting its consistency and reliability across diverse examples._
>
> A: The analysis in Section 3.2 can prove that: **sentences with different genders can activate the same neurons.** Take Table 2 and 3 as an example. When projecting the neurons into unembedding space [4,5] to analyze the neurons’ interpretability, **the neurons’ top tokens are related to “man” and the last tokens are related to “woman”.** This means that the neurons save the gender bias. **If the neuron’s coefficient score is larger than zero, it is useful for increasing the probability of “man” and decreasing the probability of “woman”. If the coefficient score is smaller than zero, it is useful for decreasing the probability of “man” and increasing the probability of “woman”.**
>
> Furthermore, **our experiments are done in the commonly used gender bias datasets. These datasets contain not only the profession cases, but also other gender-bias sentences.** The results in Table 4 and Table 5 can prove that our method can achieve good results even if we do not analyze all the gender-bias sentences’ neurons. And this result indicates that different gender-bias sentences can activate similar neurons. It is an advantage of our method that we do not need to locate the neurons of all the gender-bias sentences. As a result, our method can be done with very small number of sentences and very fast speed.
>
> _Q: Will different types of gender-biased sentences activate distinct important neurons? The selected sentences focus on professions. If sentences featuring other gender-stereotyped topics, such as personality traits or colors, are used, would we observe similar results?_
>
> A: **The gender bias datasets contain other gender-stereotyped topics, such as personality traits or colors. The experimental results on these datasets can prove that the bias is reduced.** When only use 10 sentences, the result is already good, indicating that this is a promising way to reduce the gender bias.
>
> But we agree that doing analysis on other cases can help understand the mechanism better. We analyze the cases "The strong person is a" -> "man", "The beautiful person is a" -> "woman". We find that **most of the activated neurons are the same with the important neurons identified by the professions.** For instance, Table 1's attention neuron $A^{18,7}_{54}$'s coefficient score is 0.5281 in "The beautiful person is a" -> "woman". This is not surprising because the interpretability results can prove that **this neuron contains the information about increasing "woman" probability and decreasing "man" probability when the coefficient score is larger than zero.**
>
> _Q: What happens if the sentence is changed to "This woman is a ==> nurse"?_
>
> A: In this case, **the shallow FFN neurons and the attention neurons are similar.** For example, the shallow FFN neuron $F_{2026}^4$ is activated (coefficient: 0.0108) by the word "woman", and added into this position's residual stream. Then the attention neuron $A^{18,7}_{83}$ (coefficient: 0.0361) is activated by the woman position's hidden states. The deep FFN neurons are different, because these identified neurons contain the information about the word "nurse", rather than "woman".
>
> [1] Locating and Editing Factual Associations in GPT
>
> [2] Dissecting Recall of Factual Associations in Auto-Regressive Language Models
>
> [3] A trip towards fairness: Bias and de-biasing in large language models
>
> [4] Transformer Feed-Forward Layers Build Predictions by Promoting Concepts in the Vocabulary Space
>
> [5] Analyzing Transformers in Embedding Space

---

### Meta-Review · Area_Chair_DQDx · 2024-12-21

**Metareview:**

This paper proposes to mitigate the gender bias in LLMs through interpretable model editing. Interpretable model editing is a process of locating the key neurons in LLMs related to the gender bias, and then editing the relevant neurons by adjusting their coefficients. Reviewers agree that the studied issues are crucial, and the method shows advantages by mitigating bias while maintaining the overall model performance. However, reviewers also raise several important concerns, such as comparison with existing literature (iAJp), limited/incremental technical contribution (iAJp), outdated models (W11j), and significance of identified neurons under different inputs (jVvc, W11j). There are a series of discussions between the authors and reviewers during the rebuttal phase. While the authors resolve some of the concerns, the reviewer still expresses major concern regarding lacking comparison with prior works. Although some of the discussed literature may not be comparable due to their computation costs, there are also other existing works that study the gender bias in LLM and should be discussed and compared to demonstrate the effectiveness of the proposed method over existing literature. Considering the reviewers’ opinions and the unresolved concerns, the AC recommends making further improvements before the paper can be accepted.

**Additional Comments On Reviewer Discussion:**

Overall, there are active discussions between the authors and some of the reviewers.

Reviewer jVvc’s main concern is how the active neurons will change if the input sentences are different. The authors refer to the analysis in the paper, showing that two different sentences will activate similar neurons. This concern is carefully considered when making the final decision, but less weighed due to the reviewer not engaging in further discussion. The rebuttal is helpful, but a more direct rebuttal is to conduct experiments on large amounts of sentences, analyzing if these sentences activate similar neurons on a large scale.

Reviewer rUmF focuses on whether the method requires careful hyperparameter search during fine-tuning. The authors’ rebuttal shows that hyperparameter search is not required in the main algorithm. As reviewer rUmF does not raise further concerns, this is considered resolved when making the final decision.

Reviewer iAjp lists a series of potential weaknesses and has active discussions with the author, and some of the concerns are not sufficiently resolved. The most significant weaknesses are the incremental technical contribution and the failure to discuss and compare with existing literature. These are important concerns when making the final decision. The author argues that the innovation lies in applying existing techniques to an unsolved problem (gender bias in LLM), and the two example existing works mentioned by iAjp cannot be compared due to the computational cost. Although some of the discussed literature may not be comparable due to their computation costs, there are also other existing works that study gender bias in LLM (e.g., prompt-based methods) and should be discussed and compared.

Reviewer W11j’s main concerns include the outdated models used in experiments and the activation of neurons across different prompts (similar to jVvc). The authors’ rebuttal shows that the latest LLM still suffers from gender bias problems, while reviewer W11j still expresses concerns regarding the significance of the work with growing LLMs and the impact of different input settings. These concerns are also taken into account when making the final decision.

Reviewer wmNw’s main concern lies in the details of experiment settings, which are properly addressed by the authors’ rebuttal as indicated by the reviewer.

Considering all the points mentioned above, some of the concerns are not sufficiently resolved. The AC therefore recommends making further improvements before the paper can be accepted.

---

### Decision · Program_Chairs · 2025-01-22

Reject